# Design and Fabrication of a High-Frequency Single-Directional Planar Underwater Ultrasound Transducer

**DOI:** 10.3390/s19194336

**Published:** 2019-10-08

**Authors:** Qiguo Huang, Hongwei Wang, Shaohua Hao, Chao Zhong, Likun Wang

**Affiliations:** 1School of Science, Beijing Information Science and Technology University, Beijing 100192, China; hjg303@163.com (Q.H.); 15652868686@163.com (S.H.); zclovelxm@163.com (C.Z.); 2Sensing Technology Research Center, Beijing Information Science and Technology University, Beijing 100101, China; wlikun@bistu.edu.cn

**Keywords:** high-frequency, single-directional, underwater ultrasound transducer, piezoelectric ceramics, 1-3 piezoelectric composites, ANSYS finite element software

## Abstract

This paper describes the fabrication of 1-3 piezoelectric composites by using PZT5-A pure piezoelectric ceramics and the preparation of a high-frequency single-directional planar underwater ultrasound transducer by using the developed composites. First, three material models of the same size were designed and simulated by ANSYS finite element simulation software. Next, based on the simulation results, the 1-3 piezoelectric composites were developed. Finally, a high-frequency single-directional planar underwater ultrasound transducer was fabricated by encapsulating and gluing the 1-3 piezoelectric composites. The performance of the transducer was tested, and results showed that the device was characterized by single-mode operation in the working frequency band, a high transmitting voltage response, and single directivity.

## 1. Introduction

Lead zirconate titanate (PZT) piezoelectric ceramic, a functional material, has been widely used in the manufacture of underwater acoustic transducers [1,2] because of its high electromechanical coupling coefficient, wide dielectric constant range, low mechanical loss, easy fabrication and low cost. However, due to the high acoustic impedance of PZT pure piezoelectric ceramics, their transverse coupling in thickness mode resonance is large, the bandwidth is narrow, and the transducer made of it can hardly to match with water and human body. Relevant researchers have put forward the idea of compounding PZT pure piezoelectric ceramics with polymers and developing piezoelectric composites. Piezoelectric composites are composed of PZT piezoelectric ceramic phase materials and polymer phase materials with a certain connectivity, volume, weight, or spatial geometric distribution. Common composites include 0-0, 0-1, 0-2, 0-3, 1-1, 1-2, 1-3, 2-2, 2-3, and 3-3. Of these, 1-3 piezoelectric composites are the most widely studied and applied materials. Research on 1-3 piezoelectric composites has mainly concentrated on their theoretical analysis, preparation, and application.

### 1.1. Theoretical analysis of 1-3 Piezoelectric Composites

In the late 1970s, Newnham at Pennsylvania State University proposed a theoretical model of series-parallel connection of 1-3 piezoelectric composites based on the difference between the direction of the PZT piezoelectric ceramic (or polymer) phase and the applied electric field in composite materials [3,4]. The equivalent performance parameters of 1-3 piezoelectric composites in different connection modes were also calculated. As shown in Figure 1, two connection modes of 1-3 piezoelectric composites have been developed; in this figure, E stands for the electric field. When the PZT piezoelectric ceramic (or polymer) phase is perpendicular to the direction of the electric field, the series mode is formed; when the PZT piezoelectric ceramic (or polymer) phase is parallel to the direction of the electric field, the parallel mode is formed.

In the 1980s, a theoretical model of 1-3 piezoelectric composites featuring periodically arranged PZT columns was established by Auld et al. at Stanford University (CA, USA), and the transverse structure of this theoretical model was analyzed [5]. A uniform model of 1-3 piezoelectric composites applied to thickness vibration transducers was first proposed by Smith (Philips Laboratory, New York, NY, USA), so the Smith composite theory was established by them, and based on this theory, they have developed a 1-3 piezoelectric composite transducer [6,7]. The influence of the volume fraction of different PZT piezoelectric ceramic phases on the parameters of 1-3 piezoelectric composites was analyzed in detail, and the formulas for calculating these parameters were deduced. His experimental results showed that the piezoelectric coefficient, dielectric coefficient, and acoustic impedance of a composite increases linearly with the volume percentage of PZT. In addition, when the volume fraction of PZT is between 20% and 80%, the electromechanical coupling coefficient of the 1-3 piezoelectric composites is the largest.

### 1.2. Preparation of 1-3 Piezoelectric Composites

In the early 21st century, Gentilma et al. [8] prepared 1-3 piezoelectric composites used in underwater acoustic transducers by the injection molding method. Potong et al. [9] (Chiang Mai University) studied the ferroelectric properties of 1-3 piezoelectric cement-based composites based on the permutation casting method; the authors found that, the residual polarization strength of PZT composites could reach 10 μC/cm^2^ at 50 Hz when the volume fraction of the PZT piezoelectric ceramic phase reaches 60%. Further study of piezoelectric composites allowed V Y Topolov et al. [10] of Russia to prepare 1-0-3 piezoelectric composites based on 1-3 piezoelectric composites. The scholars also determined that the piezoelectric sensitivity of 1-0-3 piezoelectric composites first increases and then decreases with increasing volume fraction of the piezoelectric phase.

### 1.3. Applications of 1-3 Piezoelectric Composites

The application of 1-3 piezoelectric composites to acoustics has been studied by Guatieri [11] and Solal et al. [12]. Relevant studies include hydrophones with low-frequency characteristics, ultrasound medical diagnostic transducers with high-frequency characteristics, and nondestructive detection transducers. Research results generally show that 1-3 piezoelectric composites have broad application prospects in nondestructive testing, acoustics, and medicine, among others. Rossetti [13] and Bent et al. [14] extensively studied the application of piezoelectric composites with interdigital electrodes to actuators, sensors, and other fields, and their research results showed that 1-3 piezoelectric composites have excellent actuation and sensing ability. Thus, these materials may replace traditional actuators and sensors in the future. NASA and Plessis et al. [15] at Colorado University studied the application of 1-3 piezoelectric composites to active rotors in aerospace and discovered that these composites show excellent performance in terms of power transfer and stability in active rotors.

The above research shows that current studies of 1-3 piezoelectric composites mainly focus on their theoretical model, preparation process, and applications. This paper is based on the research findings of relevant scholars. A 1-3 piezoelectric composite material is selected to study a high-frequency single-directional planar ultrasonic transducer with a resonant frequency of approximately 150 kHz. The design and fabrication process from a PZT pure piezoelectric ceramic sheet to a transducer was successfully realized.

## 2. Design, Simulation, and Calculation of the Model Structure by ANSYS Software

ANSYS finite element simulation software can accurately model, simulate, and calculate 1-3 piezoelectric composites in different physical fields [16,17]. Figure 2 shows the flow chart of ANSYS finite element simulation software for transducer simulation calculation. 

### 2.1. ANSYS Software for Model Structure Design

Piezoelectric ceramics are active devices. The size design of piezoelectric ceramics determines their performance. As the thickness vibration mode of piezoelectric ceramics is adopted into the design of the transducer, a PZT piezoelectric ceramic (or polymer) phase parallel to the direction of the applied electric field is adopted. Finite element simulation of piezoelectric ceramics of different thicknesses was carried out prior to the experiments, and the curve of resonant frequency as a function of ceramic thickness is plotted in Figure 3.

As can be seen in Figure 3, the resonant frequency decreases with increasing ceramic thickness. A piezoelectric ceramic sheet with a thickness of 10 mm was selected in this transducer design to achieve a transducer resonant frequency of approximately 150 kHz. The size of high-frequency ultrasonic transducers usually ranges from tens to hundreds of millimeters. Therefore, a 1-3 piezoelectric composite model with a length of 100 mm, width of 100 mm, and thickness of 10 mm was designed.

According to research on the volume fraction of PZT piezoelectric ceramics, a 1-3 piezoelectric composite with a PZT piezoelectric ceramics volume fraction near 50% was selected for this transducer. The model of a 1-3 piezoelectric composite is shown in Figure 4. In this model, the width of the polymer phase is 0.28 mm, the width of the piezoelectric ceramic columns is 1.44 mm, and the height is 10 mm. The volume ratio of PZT piezoelectric ceramic pillars to 1-3 piezoelectric composites is 51.84%.

### 2.2. ANSYS Software for the Simulation and Calculation of the Model Structure

Because the model of the 1-3 type composite contains a two-phase material, a large calculation cost is encountered during the simulation. One of the elements of 1-3 type piezoelectric composites is thus selected to simplify the calculations. Moreover, symmetrical boundary conditions [18] must be added around the element. The simulated structure is shown in Figure 5. Figure 5a reveals a three-dimensional simulation of the 1-3 piezoelectric composite. In this model, the middle cube represents a PZT piezoelectric ceramic column wrapped by a polymer. Figure 5b shows the grid structure that must be partitioned during simulation.

In this experiment, two types of 1-3 piezoelectric composites and PZT5-A pure piezoelectric ceramics were simulated by the finite element method. The ceramic phase of the 1-3 piezoelectric composite was PZT5-A pure piezoelectric ceramic and its polymer phase was epoxy resin 618 or silicone rubber. Their polarization direction was thickness direction. The conductivity modulus G curves were obtained by simulation, as shown in Figure 6. The simulation results showed that the resonant frequency of the PZT-5A pure piezoelectric ceramic is 196.5 kHz, that of the 1-3 piezoelectric composite filled with epoxy resin 618 is 145.5 kHz, and that of silicone rubber is 136 kHz. The resonant frequency of the 1-3 piezoelectric composite with silicone rubber is lower than that of the composite with epoxy resin 618.

## 3. Fabrication and Performance Testing of the 1-3 Piezoelectric Composites

### 3.1. Fabrication of the 1-3 Piezoelectric Composite

Several methods are commonly used to develop 1-3 piezoelectric composites; these methods include cutting and filling [19,20], permutation casting [21], laser or ultrasonic cutting [22,23,24], and injection molding [8]. In this experiment, the 1-3 piezoelectric composite was fabricated by the forward and backward cutting filling method. The fabrication process is shown in Figure 7.

According to the design of the 1-3 piezoelectric composite model and the results of ANSYS simulation analysis, a pure piezoelectric ceramic sheet with a length of 100 mm, width of 100 mm, and thickness of 10 mm was selected to fabricate the 1-3 piezoelectric composite. In practical applications, 1-3 piezoelectric composites are prone to deformation after solidification due to the softness of the silicone rubber. Thus, epoxy resin 618 was applied as the polymer phase to prepare the composite.

The fabrication of the 1-3 piezoelectric composite start with cutting along the X and Y directions of a pure piezoelectric ceramic by a precision cutting machine and filling epoxy resin 618. Then, reverse cutting and gluing with the same method. Finally, the electrodes are plated on the upper and lower surfaces of the composite with silver slurry.

The advantage of using the cutting/filling method is that the size of the piezoelectric ceramic columns in the 1-3 piezoelectric composite can be flexibly controlled by changing the cutting step. This method is very simple, easy to operate and control, and the test results of the finished products are excellent. Its disadvantage, however, is that the cut piezoelectric ceramic cannot be recycled, resulting in the wastage of raw materials. Figure 8 shows a 1-3 piezoelectric composite sample prepared in this experiment.

### 3.2. Performance Testing of the 1-3 Piezoelectric Composite

The performance of the 1-3 piezoelectric composite was tested by Agilent 4294A impedance analyzer in the laboratory. As shown in Table 1, the resonant frequency of the 1-3 piezoelectric composite is 151 kHz, its bandwidth is 1.71 kHz, its sound velocity is 3940 m/s, its acoustic impedance is 17.47 Mray1, its conductivity is 104.6 mS, its electromechanical coupling coefficient is 0.68, and its mechanical quality factor is 88.18.

The conductivity G curves of the 1-3 piezoelectric composite sensing elements were plotted by using an impedance analyzer. As shown in Figure 9, the highest point of the curve corresponds to the resonant frequency of the 1-3 piezoelectric composite sensor (151 kHz). The result obtained from ANSYS finite element simulation software is 145.5 kHz, which means the error rate 3.6%. These findings indicate that the simulation and test results are basically in agreement.

## 4. Fabrication and Performance Test of the High-Frequency Single-Directional Planar Underwater Ultrasound Transducer

### 4.1. Fabrication of the High-Frequency Single-Directional Planar Underwater Ultrasound Transducer

Prior to the design and fabrication of the underwater ultrasound transducer, the mold used to produce the transducer must be obtained. The size of the mold should be based on the size requirements of the transducer. The basic structure of the underwater ultrasound transducer can be divided into a matching layer [25,26], sensitive components, electrode lead, rigid foam, and metal cover plate. 

The matching layer is also called the waterproof sound transmission layer; this layer prevents leakage and matches with the water medium to ensure that ultrasound waves radiate through this layer and into the water. Polyurethane is selected as the waterproof and sound transmission layer material in this work. The density, sound velocity, and characteristic impedance of polyurethane are close to those of water and can thus match water better. Polyurethane is a suitable material for fabricating waterproof and sound transmission layers. The thickness of the waterproof sound transmission layer must be strictly set to allow ultrasonic waves to completely transmit through the polyurethane layer because the interface between the polyurethane layer and water can reflect the ultrasonic waves generated by the vibration of the sensitive element. The design theory of this layer is based on quarter-wavelength matched total transmission theory [27,28]. The thickness of the polyurethane layer can be calculated according to Equation (1).
(1)$$t=\frac{1}{4}\lambda$$

In Equation (1), t is the thickness of the matching layer and λ is the wavelength of the sound wave in the matching layer. Because the resonant frequency of the 1-3 piezoelectric composite sensor is 151 kHz, the frequency of the ultrasonic wave is 151 kHz at the resonant frequency, and the propagation speed of the ultrasonic wave in polyurethane is 1500 m/s. Therefore, the wavelength of the ultrasonic wave in the polyurethane material is about 10 mm; the thickness of the polyurethane layer is 2.5 mm.

Rigid foam is adopted as the backing material of the transducer; this component mainly acts as a fixed sensing element and absorbs vibrations at the back of the sensing element. The density of the rigid foam is very low to reduce the weight of the transducer. The dense porous structure of this foam can effectively absorb acoustic radiation from the back of the sensitive components. The dimensions of this foam are identical to those of sensitive components.

The design of the metal cover should consider the connection with the electrode wire and the testing position of the transducer. The metal cover is made of stainless steel and of the same length and width as the sensitive components; moreover, it is thinner than the sensitive components.

The assembly and pasting method of each component of the planar underwater ultrasound transducer is shown in Figure 10. Figure 11 provides an exploded schematic of the transducer.

The completed transducer is shown in Figure 12. The conductance G curve of the transducer is tested by an impedance analyzer, and the results are shown in Figure 13. According to Figure 13, the red curve represents the conductivity G curve of the 1-3 piezoelectric composite, the blue curve represents the test results of the transducer in air, and the green curve represents the test results of the transducer in water. The test curves show that the resonant frequency of the transducer is identical to that of the 1-3 piezoelectric composite sensor, which is 151 kHz. Because of the load provided by the polyurethane layer, the conductivity of the transducer in air is less than that of the 1-3 type piezoelectric composite. When the transducer is placed in water, the load of water is stronger than that of polyurethane, and the conductivity of the transducer is lower.

### 4.2. Performance Test of the High-Frequency Single-Directional Planar Underwater Ultrasound Transducer

The performance parameters of the transducer include transmitting sound source level, transmitting voltage response, receiving sensitivity, and directivity. These parameters must be measured in an anechoic tank. The measurement system of transducer is shown in Figure 14. The system includes a signal generator, power amplifier, amplifier filter, oscilloscope, standard hydrophone, and motion controller, among others.

The transmitting sound source level of an underwater ultrasound transducer describes the intensity of the sound signal emitted by the transducer at the transmitter; this property is used to measure the radiated sound intensity of the source in a certain direction and defined as the sound pressure level at a location *d* away from the source. The formula for calculating the transmitting sound source level is calculated according to Equation (2):(2)$$Lp=20\mathrm{lg}(U_{1}/d)-M_{1}$$

In Equation (2), Lp denotes transmitting sound source level, $U_{1}$ denotes the voltage received by a standard hydrophone, and $M_{1}$ denotes the receiving sensitivity of a standard hydrophone.

The transmitting voltage response is one of the more important electro-acoustic parameters describing the emission performance of transducers; it is defined as the ratio between the sound pressure in the acoustic field radiated by the transducer and the voltage at the input end of this transducer. The transmitting voltage response is calculated according to Equation (3):(3)$$Sv=20\mathrm{lg}(U_{1}/U_{0}\times d)-M_{1}$$

In Equation (3), Sv denotes the transmitting sound source level, $U_{1}$ denotes the voltage received by a standard hydrophone, $U_{0}$ denotes the voltage at both ends of the transducer to be measured, and $M_{1}$ denotes the receiving sensitivity of a standard hydrophone.

The standard hydrophone is positioned in front of the transducer to be measured at a distance of *d* in the laboratory when testing the transmitting sound source level and the transmitting voltage response. The transducer to be measured is fixed on the motion controller in accordance with the height of the standard hydrophone. The frequency range of scanning (approximately 70−220 kHz) and the environmental parameters of testing (as shown in Table 2) are defined in the computer software. The signal generator generates a sinusoidal signal, and the transducer is driven by the amplified sinusoidal signal of the power amplifier. At this time, the voltage $U_{0}$ loaded on both ends of the transducer is collected by the piezoelectric current sensor. After the standard hydrophone at distance *d* receives the acoustic signal, the voltage $U_{1}$ at both ends of the standard hydrophone is collected by the piezoelectric current sensor, and the waveform is displayed by the oscilloscope. 

Receiving sensitivity is an important parameter describing the receiving performance of a transducer and reflects its perception of a sound field; it is defined as the ratio of the output voltage of the transducer to the sound pressure of the transducer at the receiving terminal. The receiving sensitivity of the transducer is measured by the comparison method. First, a stable transmitting transducer is selected as the sound source to establish a sound field. Next, keeping the measurement environment unchanged, the transducer to be measured and the standard hydrophone are placed at a position *d* from the sound field in sequence to ensure that the transducer to be tested and the standard transducer accept the same sound pressure. Finally, the sensitivity of the transducer to be measured is obtained by comparing the output voltage of the transducer with that of the standard hydrophone.

Directivity is an important index describing the acoustic radiation range of transducers. In this experiment, the original test environment is kept unchanged, the transducer is fixed on the motion controller of the transmitter such that its direction is controlled by the motion controller. Next, the receiving voltage of the hydrophone is read by an oscilloscope. Then, the received voltage is converted into decibel form, and the angle at which the maximum voltage falls by 3 dB is read out; this angle is also known as the beam opening angle of the transducer.

The results of the test are shown in Figure 15. Figure 15a shows the transmitting sound source level of the transducer, while Figure 15b shows the transmitting voltage response of the transducer. Figure 15c shows the receiving sensitivity of the transducer, and Figure 15d shows the directivity of the transducer.

According to Figure 15a,b, when the resonant frequency is 151 kHz, the transducer’s transmitting sound source level and voltage response are highest, the transmitting sound source level is 220.1 dB, and the transmitting voltage response is 183.8 dB. In Figure 15c, the sensitivity of the transducer at the resonant frequency is −184.3 dB. However, when the vibration frequency of the transducer is 180 kHz, its maximum sensitivity is −174.4 dB because the frequency corresponding to the maximum sensitivity is the anti-resonant frequency of the transducer. During the directivity test, the direction of transducer was 16° ahead of the standard hydrophone. In the directional pattern shown in Figure 15d, the main lobe is sharp and the left and right side lobes are small. In addition, when the beam width of the transducer is −3 dB [29], the beam opening angle of the transducer is only 2.4°, that is to say, the directivity of the transducer is single. The test results confirm that a high-frequency single-directional planar underwater ultrasound transducer was successfully fabricated in this experiment.

## 5. Conclusions

A high-frequency single-directional planar underwater ultrasound transducer is introduced and analyzed in this paper. ANSYS finite element simulation software was used to design the model structure, and three structural models with the same size were simulated and analyzed. The resonant frequency of 1-3 piezoelectric composites could be reduced by addition of flexible polymers. Based on the simulation results, epoxy resin 618 was selected as the polymer phase for filling, and the forward and backward cutting/filling method was employed to realize the design and fabrication of 1-3 piezoelectric composites. The high-frequency single-directional planar underwater ultrasound transducer was successfully fabricated. The performance of the transducer was tested in water and air, and results showed that the resonant frequency of the transducer is identical to that of the 1-3 piezoelectric composite (151 kHz). When the resonant frequency of the transducer is 151 kHz, its transmitting sound source level is as high as 220.1 dB, its transmitting voltage response is as high as 183.8 dB, and its receiving sensitivity is −184.3 dB. Moreover, the directivity angle of the transducer is only 2.4° when the beam width is −3 dB. The fabrication results of the transducer are basically consistent with the simulation results. The transducer can be used to receive and transmit underwater ultrasound communication signals; it can also be applied to underwater ranging, target recognition, and unmanned vehicles.

## Figures and Tables

**Figure 1 sensors-19-04336-f001:**
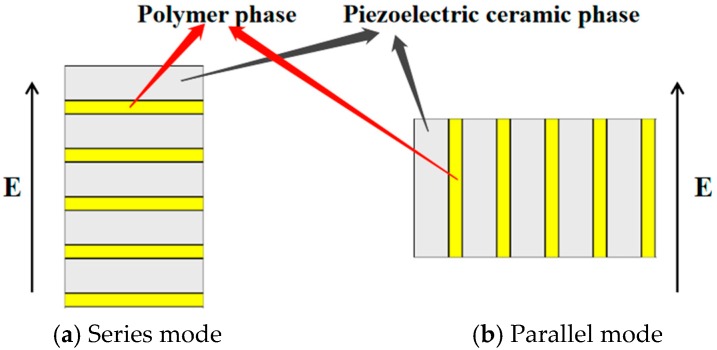
Two connection modes of 1-3 piezoelectric composites.

**Figure 2 sensors-19-04336-f002:**
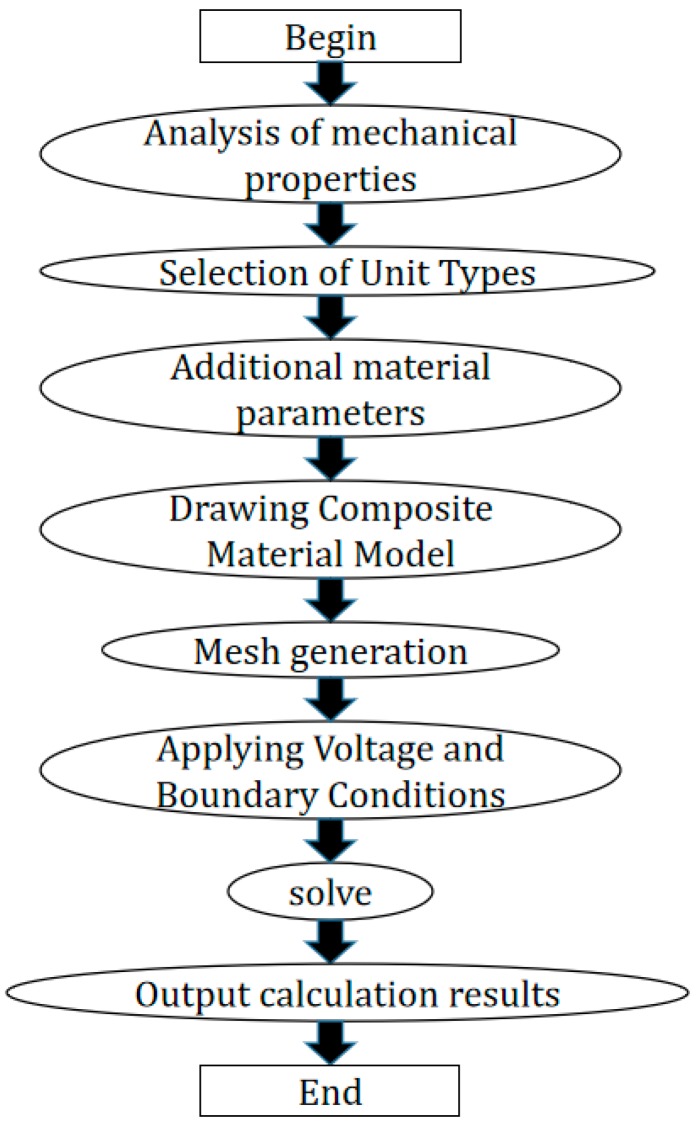
Simulation flow chart of ANSYS finite element software.

**Figure 3 sensors-19-04336-f003:**
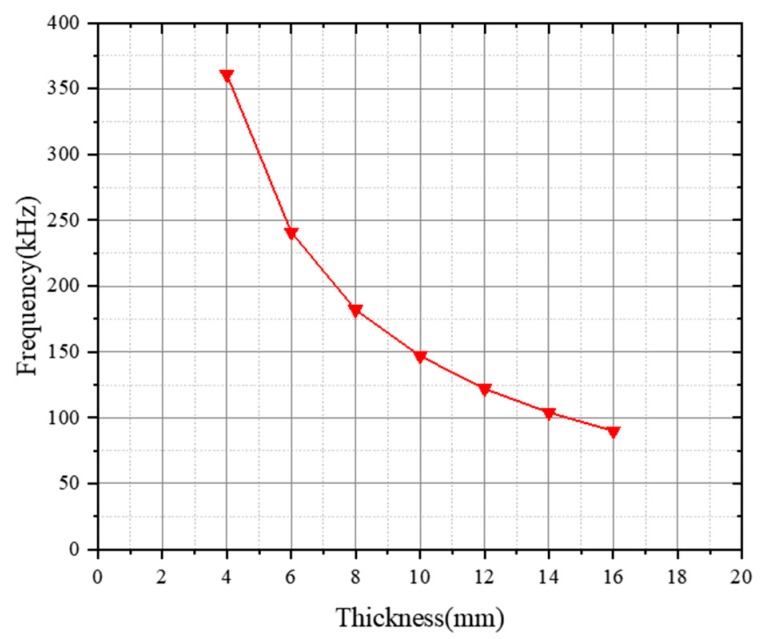
Curve of resonant frequency as a function of ceramic thickness.

**Figure 4 sensors-19-04336-f004:**
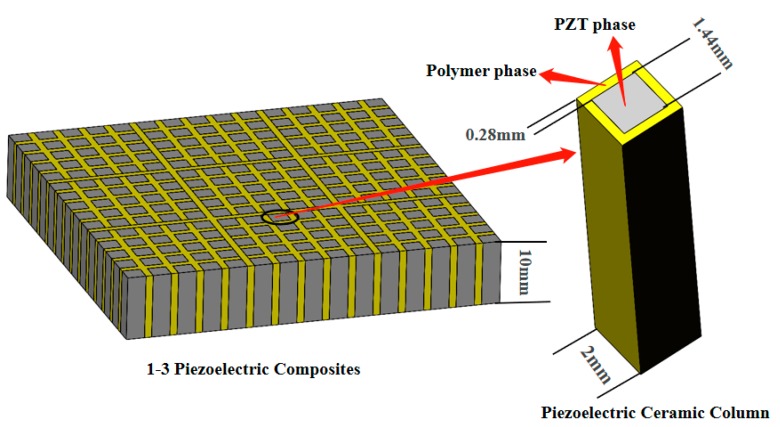
Model structure of a 1-3 piezoelectric composite and sizes of the PZT and polymer phases.

**Figure 5 sensors-19-04336-f005:**
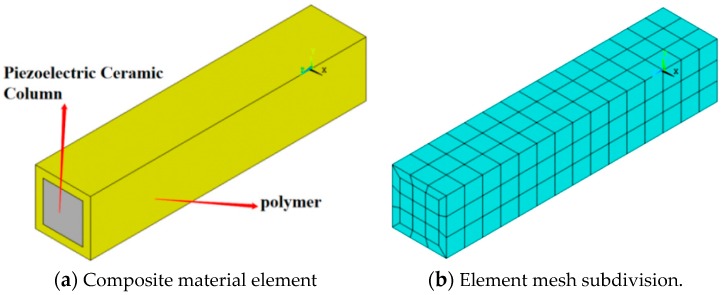
Simulated structural diagram of a 1-3 piezoelectric composite.

**Figure 6 sensors-19-04336-f006:**
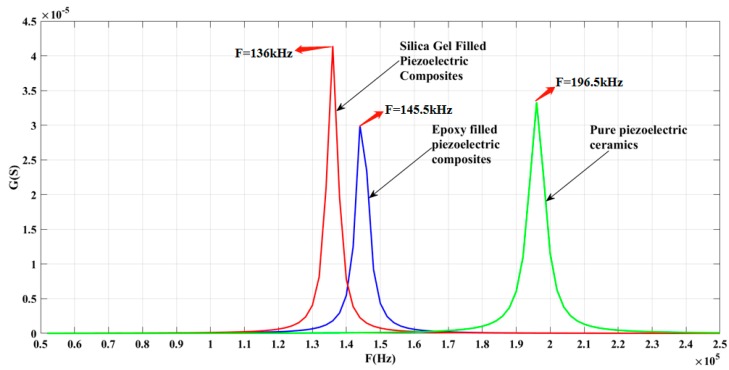
Simulation results of pure piezoelectric ceramics and two polymer-added 1-3 piezoelectric composites.

**Figure 7 sensors-19-04336-f007:**
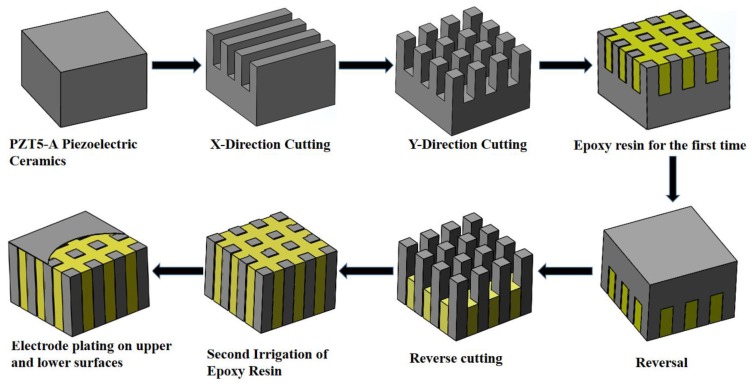
Fabrication process of the piezoelectric composites.

**Figure 8 sensors-19-04336-f008:**
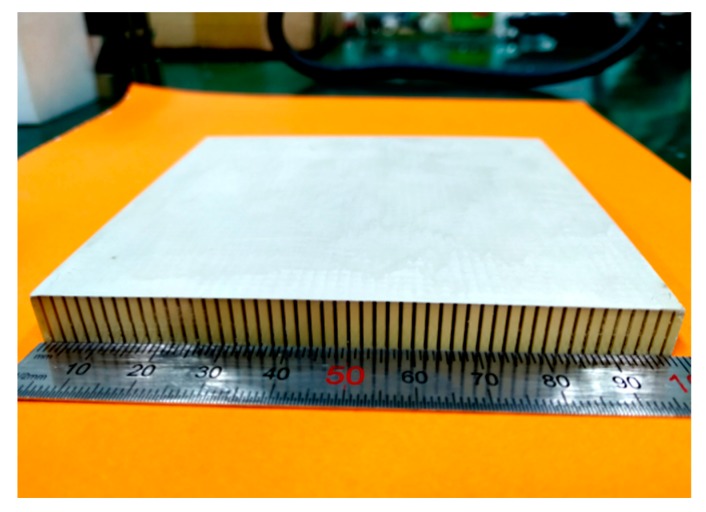
Photograph of the 1-3 piezoelectric composites.

**Figure 9 sensors-19-04336-f009:**
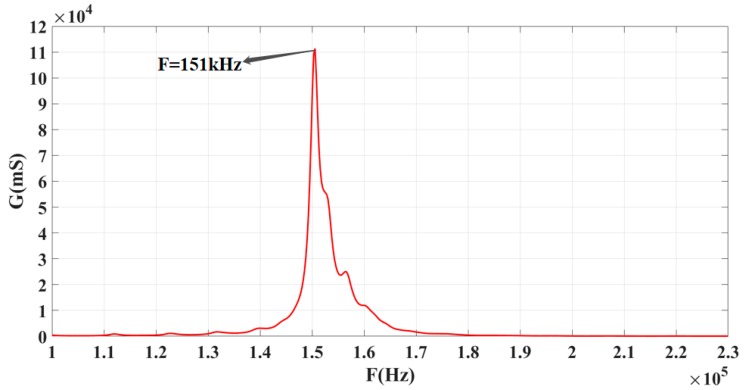
Conductivity G curve of the piezoelectric composite sensor.

**Figure 10 sensors-19-04336-f010:**
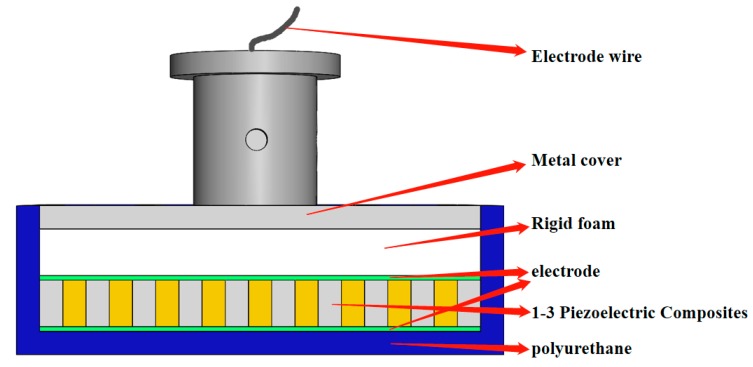
Structural diagram of the planar transducer.

**Figure 11 sensors-19-04336-f011:**
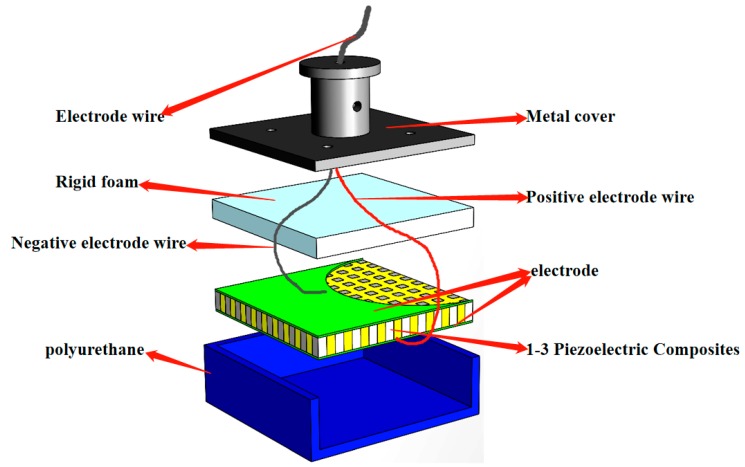
Exploded schematic of the planar transducer.

**Figure 12 sensors-19-04336-f012:**
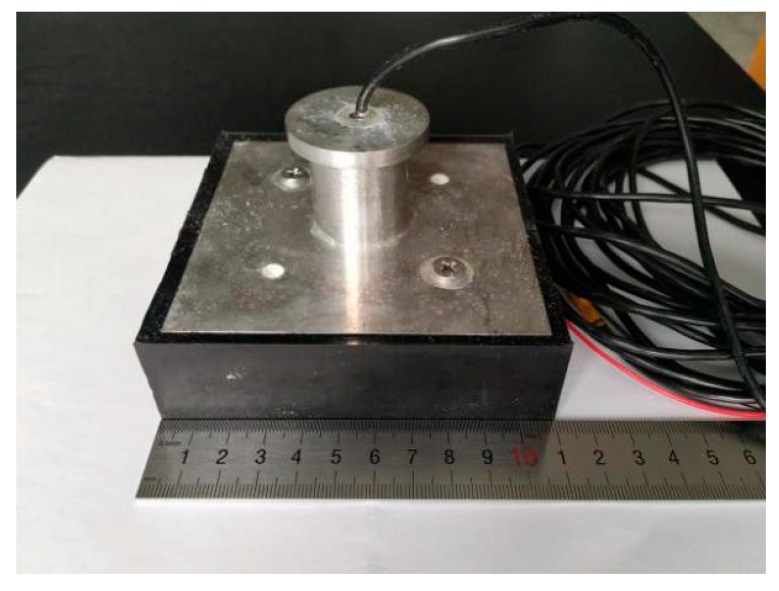
Physical diagram of the underwater acoustic transducer.

**Figure 13 sensors-19-04336-f013:**
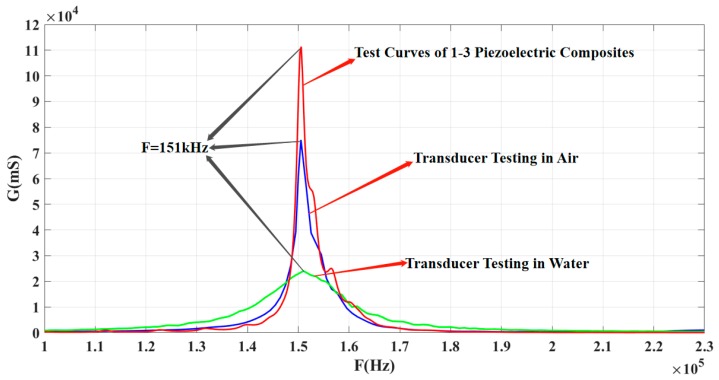
Conductivity G curves of the piezoelectric composite and transducer.

**Figure 14 sensors-19-04336-f014:**
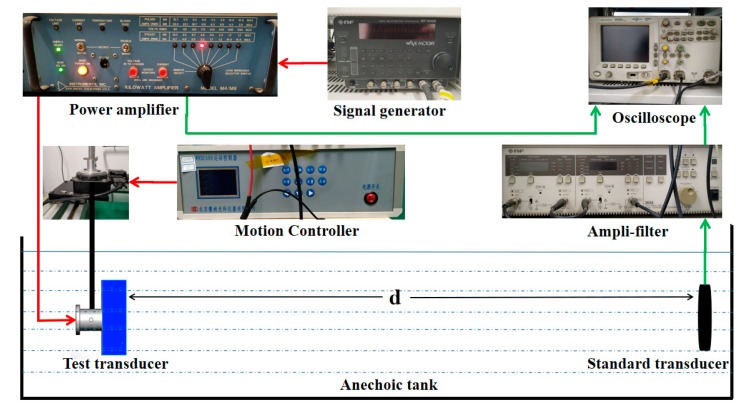
Test system for the transducer.

**Figure 15 sensors-19-04336-f015:**
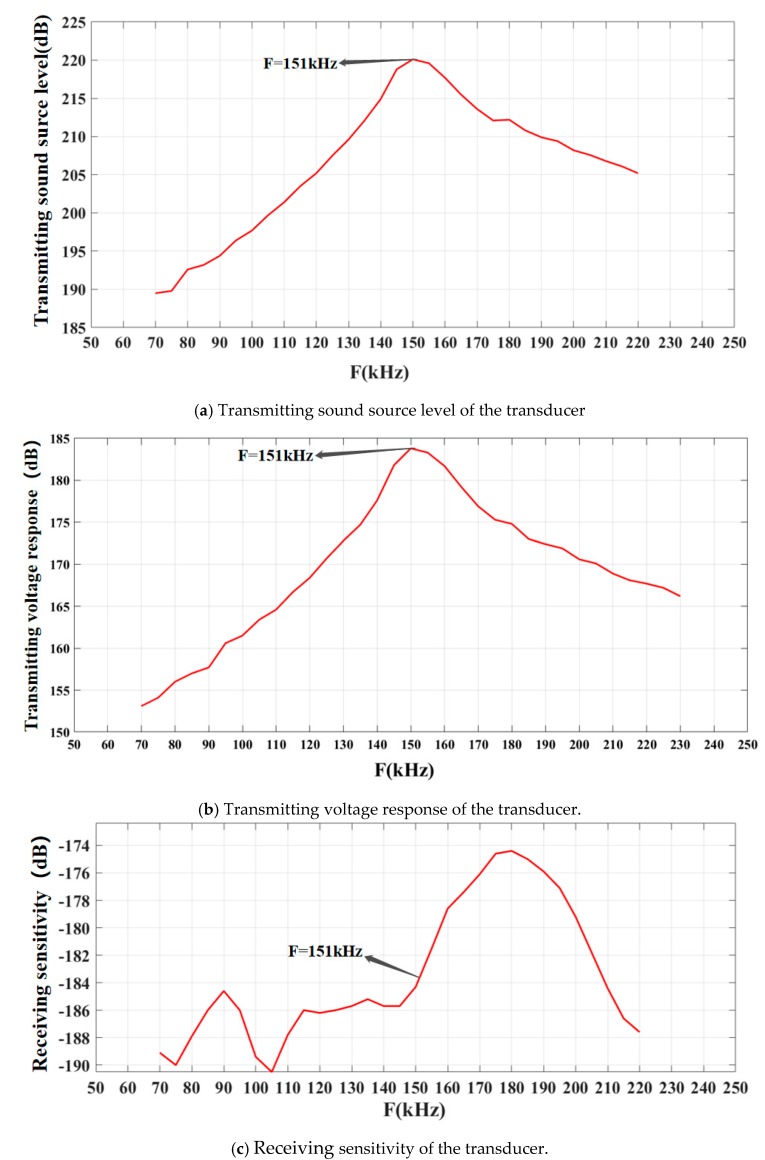
Performance parameters of the transducer.

**Table 1 sensors-19-04336-t001:** Performance parameters of the 1-3 piezoelectric composites.

Length (mm)	Width (mm)	Thickness (mm)	Quality (g)	Density (kg/m^3^)	Bandwidth (kHz)
100	100	10	443.52	4435.2	1.71
**Sound Velocity** **(m/s)**	**Acoustic Impedance** **(Mray1)**	**Resonant Frequency** **(kHz)**	**Conductance** **(mS)**	**Electromechanical Coupling Coefficient**	**Mechanical Quality Factor**
3940	17.47	151.00	104.6	0.68	88.18

**Table 2 sensors-19-04336-t002:** The test environment of the transducers.

Ambient Temperature	Cable Length	Depth of Water Inflow	Water Temperature	Insulation Resistance	Direct Capacitance	Test Distance
21.0 °C	8.0 m	0.4 m	18.0 °C	500 MΩ	9000 pF	0.87 m

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
