# Peer review of "Design and Fabrication of a High-Frequency Single-Directional Planar Underwater Ultrasound Transducer"

_sensors, 2019, doi:10.3390/s19194336_

Round 1
Reviewer 1 Report
In this manuscript, thie main contents of this paper are the following two aspects. One aspect is to fabricate 1-3 piezoelectric composites by using PZT5-A pure piezoelectric ceramics. Another aspect is to fabricate high-frequency single-directional planar underwater acoustic transducer by using the developed 1-3 piezoelectric composites. The test results showed that the transducer had the characteristics of single mode in the working frequency band, high emission voltage response and single directivity. . However, there are some problems need to be discussed. Specific comments are as follows:
1) We can't find the relationship Equations (1) ~(10) and Section 2, 3. If Equations (1) ~(10) are only references, they should describe simply.
2) 1-3 piezoelectric composites have studied for several decades. In introduction, some published papers about 1-3 piezoelectric composites and their researches in Underwater Acoustic Transducer, especially the problems in these researches.
3) Where are the application field of 1-3 piezoelectric composites Underwater Acoustic Transducer?
4) In this paper, after each Equation, the authors use "In the equation" or "In the above equation. However, in most papers, "where" was used.
Author Response
Thank you for your sincere comments and suggestions. Your proposal has helped me a lot. I have revised the article and sent it to the editor. I have responded to your proposal in the submitted document.

Reviewer 2 Report
Nice paper on 1-3 composite and comparison to analytical model and finite element - generally good work. Check Kg, should be kg since K=Kelvin. English can be generally improved, otherwise good.
Author Response

(The authors gave the same response as above.)

Reviewer 3 Report
Review of manuscript Sensors-569594
The authors present the design, construction and test of an ultrasonic transducer that operates at 151 kHz and describe its performance. Experimental work has been carried out to verify the design.
However, I have a few objections to the publication of the manuscript.
1. First objection is that Section 1 presents a theoretical analysis and it is not clear if that analysis was already presented in papers cited [10] and [12]. Then, in Section 2, the analysis of composites behavior and the transducer design are addressed with the help of ANSYS. I do not find any connection between the theoretical analysis of Section 1 and the simulation process of Section 2. Neither any clue is found on how the single piezoelectric-polymer composite column shown in Fig. 3 has been designed.
2. Equations (1) to (10) are not easy to read, because the symbols are neither presented on time nor well described. In lines 53-54 one can read “The connection mode of 1-3 piezoelectric composites studied in this experiment is parallel connection mode.” But Eqs. (1)-(2), as mentioned in line 55, are for the series connection mode, which makes the text unclear for the reader. I would like also to point that maybe there is a typographic mistake in one superscript in one term of the numerator of Eq. (1).
3. The test set description in Table 4 (not Table 4.2, as said in line 209) is insufficient. The description of the measured magnitudes shown in Figs. 12.a and 12.b, is also insufficient. For instance: The transducer has been tested in the transmitter or the receiver side or both? If in the transmitter side, what is the driver voltage or the electric power delivered to the transducer to obtain the source level shown in Fig.12.a? And so on.
4. References are not consistent: formats are not homogeneous; names, first names and titles follow different orders. Documents difficult to reach, like Dissertations or papers in University Journals, should be avoided. In addition, I find very convenient to provide the publication DOI when available.
5. And a last objection is how careful the manuscript edition is. First of all, this manuscript needs a thorough English language review. There is also an important issue on how ideas are presented and, mainly, how they are linked, as already mentioned in objection 1. Finally, authors could pay more attention to details. Some examples have already been mentioned above in previous comments, but I will mention a few more:
* magnitudes h and \beta in Eq. (8) are not defined.
* in lines 86-87, one reads that “E denotes electric field strength, D denotes electric displacement component”, but in Eqs. (5)-(10), E and D are two superscripts and their meaning is different.
* in line 165 it is said that the acoustic impedance is 17.47 but units are not. We need to read Table 3 to see that units are Mrayl.
* nowhere is said that the operating frequency of the transducer is 151 kHz. In lines 187-188, when the thickness of the matching layer is calculated, it is said “Because the wavelength of acoustic wave in epoxy resin is about 8 mm”, without any mention to frequency. By the way, the title of the manuscript could say Ultrasonic, instead of Acoustic.
* nowhere is said that results in Fig. 8 are measured results.
And many more. It is not fair that a manuscript is not carefully edited but authors expect the reviewers to do that task.
Author Response

(The authors gave the same response as above.)
